Integrated analysis of ATAC-seq and RNA-seq reveals ADSCP2 regulates oxidative phosphorylation pathway in hypertrophic scar fibroblasts

Li Qian 1
Quan Zhe 2
Chen Ling 1
Yin Yiliang 3
Chen Xin 4 kaisanju@163.com
Li Jingyun 5 drlijingyun@163.com
1 Department of Plastic & Cosmetic Surgery, Women’s Hospital of Nanjing Medical University (Nanjing Women and Children’s Healthcare Hospital) , Nanjing , China
2 Department of Dermatology, Tongren Hospital, Shanghai Jiao Tong University School of Medicine , Shanghai , China
3 Department of Burns and Plastic Surgery, Yancheng No. 1 People’s Hospital, Affiliated Hospital of Medical School, Nanjing University, Yancheng, China. The First People’s Hospital of Yancheng , Yancheng , China
4 Department of Cardiology, Nanjing Drum Tower Hospital, Affiliated Hospital of Medical School, Nanjing University , Nanjing , China
5 Nanjing Women and Children’s Healthcare Institute, Women’s Hospital of Nanjing Medical University (Nanjing Women and Children’s Healthcare Hospital) , Nanjing , China
Nunes-da-Fonseca Rodrigo
Electronic publication date: 2025 Jan 31
Publication date: 2025
Volume: 13
Electronic Location ID: e18902
Received 2024 Aug 9; Accepted 2025 Jan 3
Copyright: © 2025 Li et al.
Copyright year: 2025
Copyright holder: Li et al.
License: This is an open access article distributed under the terms of the Creative Commons Attribution License, which permits unrestricted use, distribution, reproduction and adaptation in any medium and for any purpose provided that it is properly attributed. For attribution, the original author(s), title, publication source (PeerJ) and either DOI or URL of the article must be cited.
License URL: https://creativecommons.org/licenses/by/4.0/

Keywords: Hypertrophic scar fibroblasts, ADSCP2, ATAC-seq, RNA-seq, OXPHOS

Funding: National Natural Science Foundation of China 82072185, 82002041 Jiangsu Provincial Key Research and Development Program BE2019619 Nanjing Medical Science and Technique Development Foundation YKK21164 This work was supported by the National Natural Science Foundation of China (82072185, 82002041), the Jiangsu Provincial Key Research and Development Program (BE2019619), and the Nanjing Medical Science and Technique Development Foundation (YKK21164). The funders had no role in study design, data collection and analysis, decision to publish, or preparation of the manuscript.

==============================
The primary effector cells involved in the formation of hypertrophic scars are fibroblasts. A potential peptide, ADSCP2 (adipose-derived stem cell peptide 2, the peptide fragment of ALCAM protein), derived from adipose-derived stem cell-conditioned medium, has been identified as having the potential to mitigate hypertrophic scar formation by targeting pyruvate carboxylase. However, the underlying mechanisms remain incompletely understood. Whether ADSCP2 attenuates hypertrophic scar fibrosis at the transcription level remains unclear. Consequently, this study sought to elucidate the potential mechanism associated with ADSCP2 by examining genome-wide transcriptional alterations and changes in chromatin accessibility in fibroblasts. This was achieved through the integrated analysis of assay for transposase accessible chromatin using sequencing (ATAC-seq) and RNA sequencing (RNA-seq). In the ADSCP2 treatment group, ATAC-seq identified a total of 7,805 differential peaks associated with 3,176 genes. RNA-seq analysis revealed 345 upregulated and 399 downregulated transcripts in the same group. A combined Kyoto Encyclopedia of Genes and Genomes (KEGG) pathway enrichment analysis of both downregulated genes and close-ACRs (accessible chromatin regions) genes within the ADSCP2 treatment group indicated regulation of the oxidative phosphorylation pathway (OXPHOS) by ADSCP2. The amalgamation of ATAC-seq and RNA-seq data elucidates that two OXPHOS associated genes, namely COX6B1 (cytochrome c oxidase subunit 6B1) and NDUFA1 (NADH dehydrogenase (ubiquinone) alpha subcomplex-1), demonstrate significant downregulation in the presence of ADSCP2. Further analysis using the integrative genomics viewer indicates that the promoter regions of both COX6B1 and NDUFA1 exhibit a higher degree of closure in the ADSCP2 treatment group. Quantitative PCR analysis demonstrated that ADSCP2 treatment resulted in a reduction of COX6B1 and NDUFA1 mRNA expression levels. Furthermore, cellular ATP and lactic acid concentrations were diminished in the ADSCP2-treated group. Collectively, these findings suggest potential avenues for future research into the therapeutic application of the peptide ADSCP2 in the treatment of hypertrophic scars.

Introduction

The hypertrophic scar, a fibroproliferative disorder of dermal tissue, typically manifests following surgical procedures, trauma, or burns (Chen et al., 2023). Despite considerable research efforts, the underlying mechanisms that drive hypertrophic scar formation remain elusive, rendering treatment a formidable challenge (Ogawa, 2022). In our recent study, we conducted an exhaustive evaluation of a peptide, ADSCP2 (adipose-derived stem cell peptide 2, the peptide fragment of ALCAM protein), derived from adipose-derived stem cell-conditioned medium. ADSCP2 inhibited collagen and ACTA2 mRNAs in hypertrophic scar fibroblasts. Moreover, ADSCP2 facilitated wound healing and attenuated collagen deposition in a mouse model. ADSCP2 bound with the pyruvate carboxylase (PC) protein and inhibited PC protein expression. Untargeted metabolomics identified differential metabolites in the ADSCP2-treated group. Therefore, ADSCP2 attenuated hypertrophic scar fibrosis both in vitro and in vivo (Li et al., 2023). Nevertheless, the intricate mechanisms of ADSCP2 necessitate further investigation. Whether ADSCP2 attenuates hypertrophic scar fibrosis at the transcription level remains unclear.

The management of transcriptional regulation may be accomplished by controlling the accessibility of transcription factors. The structure of chromatin is perceived to play a pivotal role in the formation of hypertrophic scars (Wang et al., 2022; Zhang et al., 2022). The accessibility of chromatin in tissue cells robustly regulates physiological activities. The ATAC-seq (assay for transposase accessible chromatin using sequencing) technology provides another method for evaluating chromatin accessibility (Grandi et al., 2022). This technology has been utilized in conjunction with RNA-seq (RNA sequencing) to investigate the transcriptional regulation network implicated in the development of systemic lupus erythematosus (Wu et al., 2023), to study cell proliferation in in vitro cultured skeletal muscle satellite cells (Ren et al., 2024), and to identify signature genes in human alpha and beta cells (Ackermann et al., 2016).

In the present study, we amalgamated gene expression (RNA-seq) and chromatin accessibility profiles (ATAC-seq) post-ADSCP2 treatment in hypertrophic scar fibroblasts. This research paves the way for the potential therapeutic application of ADSCP2 in hypertrophic scar treatment.

Materials and Methods

Ethical approval and consent to participate

Human tissue samples were procured from patients who underwent surgical procedures at the Women’s Hospital of Nanjing Medical University (Nanjing Women and Children’s Healthcare Hospital). All participants furnished written informed consent. In accordance with the Declaration of Helsinki, the study was sanctioned by the Ethics Committee of the Women’s Hospital of Nanjing Medical University (Nanjing Women and Children’s Healthcare Hospital) (Approval number: 2022KY-162).

Primary hypertrophic scar fibroblasts cell culture

Utilizing a previously established protocol (Li et al., 2023), we successfully isolated fibroblasts derived from hypertrophic scars. These hypertrophic scar-derived fibroblasts were cultured from patients (n = 3) who had not undergone any prior treatment for hypertrophic scars before their surgical excision. The cell collection process involved the use of 5% Dispase II (Sigma, Burbank, CA, USA), followed by 0.2% collagenase I (Sigma, Burbank, CA, USA). The cells were then cultured in Fibroblast Medium (Catalog #2301; ScienCell, Carlsbad, CA, USA), which comprised of basal medium, a minimal concentration of fetal bovine serum (2%, FBS, Catalog #0010; ScienCell, Carlsbad, CA, USA), and fibroblast growth supplement (FGS, Catalog #2352; ScienCell, Carlsbad, CA, USA) to aid in the preservation of the fibroblastic phenotype. The cell cultivation took place at a temperature of 37 °C in an incubator with 5% CO2 concentration. In the course of this research, fibroblasts from hypertrophic scars, specifically between passages three and seven, were utilized.

Cell treatment with ADSCP2

As previously delineated (Li et al., 2023), all peptides, with a purity exceeding 98%, were synthesized by Shanghai Science Peptide Biological Technology Co., Ltd and subsequently dissolved in distilled water (15230162; Thermo Fisher Scientific, Waltham, MA, USA). The peptide sequence for ADSCP2 was DENREKVNDQAKL, while the scrambled peptide (Sc) for ADSCP2 was NEVQADRKKELND.

ATAC-seq and sequencing data analysis

The hypertrophic scar fibroblasts were exposed to either 25 μM Sc (serving as the control) or 25 μM ADSCP2 in triplicate, after which they were consolidated into one control and one ADSCP2 sample for ATAC-seq analysis utilizing the BGI platform. ATAC-seq libraries were produced in accordance with a previously established protocol (Buenrostro et al., 2013). The process involved the isolation and purification of nuclei, which were subsequently incubated with the Tn5 transposase and supplemented with adaptors. The ATAC-seq libraries were then amplified through PCR, incorporating two distinct barcodes. Following the PCR reaction, the libraries underwent purification using Agencourt Ampure XP beads and were subsequently sequenced on a MGISEQ-2000 sequencer (BGI, Shenzhen, China) utilizing 50 bp paired-end sequencing. The raw sequencing reads were then subjected to trimming to eliminate low-quality reads and adapters, a process facilitated by fastp (version 0.22.0). The resulting clean data were aligned to the hg19 reference genome with the aid of Bowtie2 (version 2.4.2).

For the purpose of peak calling, unique reads from fragments that were less than 150 base pairs were utilized via the MACS (model-based analysis for ChIP-Seq, version: 2.1.0) algorithm. Differential peaks between samples were identified using the MAnom algorithm. The significance of each peak was determined through a Bayesian model, with significant regions being selected if the absolute value of M was greater than or equal to 1 and the p-value was less than or equal to 10−5.

Data visualization was facilitated through the use of the integrative genomics viewer (IGV 2.16.2).

RNA-seq and sequencing data analysis

In the current investigation, we used previously identified genes with the RNA-seq from hypertrophic scar fibroblasts subjected to treatment with either 25 μM Sc (control) or ADSCP2 in triplicate, followed by collection for the preparation of a transcriptome library and sequencing (Illumina NovaSeq6000, in PE150 mode), a process carried out by the ShanghaiBio Corporation (Li et al., 2023). The statistical power of this experimental design, calculated in RNASeqPower is 0.84. Here, a gene is classified as differentially expressed if it exhibits a fold change greater than 1.2 or less than 0.83, and a p-value less than 0.05 when comparing two groups. The interconnections between differential genes were scrutinized utilizing the STRING analysis tool.

Bioinformatics enrichment analysis

Differentially expressed genes (DEGs) or genes of interest were enriched using the gene ontology (GO, http://www.geneontology.org/) and the Kyoto Encyclopedia of Genes and Genomes (KEGG) database. Based on Fisher’s exact test, the top 30 or 20 enriched GO terms or pathways were listed.

Reverse transcription and quantitative polymerase chain reaction

Hypertrophic scar fibroblasts were subjected to treatment with either 25 μM Sc (control) or ADSCP2 in triplicate. Subsequently, total RNA was extracted, reverse transcribed into complementary DNA (cDNA), and amplified utilizing the SYBR Green qPCR Master Mix (Vazyme, Nanjing, China) on an ABI Vii7 PCR Detection System (ABI, CA, USA). The expression levels of COX6B1 and NDUFA1 were normalized to 18s rRNA using the 2(−ΔΔCt) method. Primer sequences for NDUFA1 were as follows: forward, 5′-GCG TAC ATC CAC AGG TTC ACT-3′; reverse, 5′-GCG CCT ATC TCT TTC CAT CAG A-3′. Primer sequences for COX6B1 were as follows: forward, 5′-CTA CAA GAC CGC CCC TTT TGA-3′; reverse, 5′-GCA GAG GGA CTG GTA CAC AC-3′.

Determination of ATP and lactic acid levels

Hypertrophic scar fibroblasts were exposed to either 25 μM Sc (serving as the control) or ADSCP2 in triplicate for a duration of 24 h. ATP concentrations were quantified utilizing the ATP Assay Kit (Catalog No. S0026; Beyotime, Shanghai, China) in accordance with the manufacturer’s protocol. Lactic acid concentrations were measured using the lactic acid LD detection kit (Catalog No. KGA7402-48; KeyGEN, Rockville, MD, USA). Protein concentrations were determined via the enhanced BCA protein assay kit (Catalog No. P0010; Beyotime, Shanghai, China), and ATP and lactic acid levels were normalized to protein content.

Results

Chromatin accessibility changes following ADSCP2 treatment

The analysis of ATAC-seq data was conducted to delineate the alterations in chromatin accessibility subsequent to ADSCP2 treatment. The control and ADSCP2 treated hypertrophic scar fibroblasts for ATAC-seq were assembled into two distinct pools. Open chromatin signals were procured through alignment, duplication removal, and peak calling (Table 1).

Table 1 Information of the ATAC-seq data.

Sample ID	Raw reads number	Clean reads number	Clean reads Q20 rate (%)	Clean reads Q30 rate (%)	Mapping rate (%)	Clean data size (bp)	
Control	113,090,848	110,031,772	98.53	95.28	98.41	5,501,588,600	
ADSCP2	91,988,726	89,591,672	98.41	94.92	98.36	4,479,583,600	
Note:

ATAC-seq, assay for transposase accessible chromatin using sequencing.

Using MACS (Zhang et al., 2008), we discerned 19,079 accessible chromatin regions (ACRs) in the control sample and 18,061 ACRs in the ADSCP2 sample (Fig. 1A). These ACRs were predominantly concentrated in the intergenic and intron regions in both control and ADSCP2 treated hypertrophic scar fibroblasts (Fig. 1B). In the control group, 50.1% and 44% of ACRs positioned at the intergenic and intron regions respectively (Fig. 1B). In the ADSCP2 group, 49.5% and 44.2% of ACRs positioned at the intergenic and intron regions respectively (Fig. 1B).

Figure 1 ATAC-seq analysis of ADSCP2 and control.

(A) Number of peaks between control and ADSCP2. ACRs: accessible chromatin regions. (B) Peak distribution on gene elements. Peaks were classified based on the location and showed in the different genome regions: intergenic, introns, downstream, upstream and exons. Up2k: upstream 2,000 bp. (C) Differential peaks distribution on gene elements. Differential binding region was obtained by software MAnorm. (D) Open regions across the entire genome. The x-axis of the main plot represents the actual chromosome sizes, the y-axis represents the reads density of open regions over the genome (E) Gene Ontology (GO) and pathway analysis of differential peaks related genes. For GO analysis on the left, x-axis represents three domains of GO while y-axis represents the gene number in every pathway and processes. For pathway analysis on the right, x-axis represents the generatio while y-axis represents the pathway. Dot sizes and colors represent the count numbers and adjusted p value.

Utilizing the MAnorm method (Shao et al., 2012), a total of 7,805 distinct ACRs were identified (Fig. 1A). The differential peaks observed between the control and ADSCP2 were primarily situated in the intergenic and intron regions (Fig. 1C). A comprehensive view of the open regions across all chromosomes was presented for the control and ADSCP2 group (Fig. 1D). A total of 3,176 genes were associated with 7,805 differential peaks, and genes exhibiting more than five differential peaks were cataloged (Fig. 1E). GO analysis indicated that the 3,176 genes associated with the 7,805 differential peaks were predominantly related to cellular processes, biological regulation, cell structure, organelle function, binding, and catalytic activity (Fig. 1F). Furthermore, the KEGG pathway analysis revealed that these differential peak genes were primarily linked with focal adhesion, calcium signaling pathway, and regulation of actin cytoskeleton (Fig. 1E).

Analysis of genes at ACRs following ADSCP2 treatment

Following the treatment with ADSCP2, the Venn analysis revealed an increase in accessibility for 640 peaks, while 1,658 peaks demonstrated reduced accessibility (Fig. 2A). Genes associated with close-ACRs were predominantly linked to the response to xenobiotic stimuli, embryonic organ development, and class II regionalization (Fig. 2B). The KEGG pathway analysis indicated that the close-ACRs genes were primarily enriched in oxidative phosphorylation, ribosome, and neuroactive ligand-receptor interaction (Fig. 2B). Conversely, genes associated with open-ACRs were predominantly linked to the development of the reproductive system, reproductive structure, and ribosome biogenesis (Fig. 2B). Analyses of KEGG pathways revealed that cytokine-cytokine receptor interaction, ribosome, and JAK-STAT signaling pathway were predominantly enriched for the open-ACRs genes, as depicted in Fig. 2B.

Figure 2 Gene Ontology (GO) and pathway analysis of accessible chromatin region (ACR) genes between ADSCP2 and control.

(A) Venn analysis of control and ADSCP2 peaks. (B) GO and KEGG pathway analysis of close and open ACR genes. ACRs: accessible chromatin regions. For GO analysis on the left, x-axis represents the count numbers while y-axis represents the processes. For KEGG pathway analysis on the right, x-axis represents the generatio while y-axis represents the pathway. Dot sizes and colors represent the count numbers and adjusted p value.

Integrated analysis of ATAC-seq and RNA-seq

In a previous study, we examined the RNA-seq data of ADSCP2, utilizing a parameter of fold change ≥2, P < 0.05, which led to the identification of 160 upregulated genes and 170 downregulated genes (Li et al., 2023). In the current investigation, we used previously identified genes, modified the parameter to fold change ≥1.2, P < 0.05, with the aim of detecting the minimal changes associated with the functions of ADSCP2. This adjustment resulted in the identification of 345 upregulated genes and 399 downregulated genes, as shown in Fig. 3A.

Figure 3 RNA-sequencing re-analysis of ADSCP2 and Control.

(A) Volcano plot. Fold change ≥1.2, P < 0.05. (B) Top 30 differential genes in the ADSCP2 group compared to the Control group. Red represents upregulation, blue represents downregulation. (C) Three domains of GO and KEGG pathway analysis of differential genes. The x-axis represents the –log of adjusted p value while y-axis represents the process or pathway. Red represents upregulation, blue represents downregulation. (D) Protein-protein interaction networks of differential genes using STRING (https://cn.string-db.org/) analysis. Lines with different colors represent known interactions, predicted interactions and others.

The study presents the top 30 upregulated and downregulated genes (Fig. 3B). An analysis of the top seven enriched GO terms, encompassing biological process, cellular component, and molecular function, revealed a predominant enrichment in cellular response to arsenic-containing substance, cytoplasmic translation, cytosolic ribosome, proteasome binding, and structural constituent of ribosome (Fig. 3C). Furthermore, an examination of the top seven enriched KEGG pathways indicated a significant enrichment in ribosome and oxidative phosphorylation (Fig. 3C). A protein-protein interaction analysis, conducted using the search tool for the retrieval of interacting genes/proteins (STRING), elucidated the intricate interaction among these genes.

An integrated analysis of ATAC-seq and RNA-seq, utilizing venn analysis, revealed that in the ADSCP2 treatment group, six genes were upregulated and exhibited increased accessibility, while 27 genes were downregulated and demonstrated decreased accessibility (Fig. 4A). The differential peaks coincided with 19 downregulated genes and 26 upregulated genes (Fig. 4A). The KEGG pathway analysis indicated that the downregulated genes were primarily associated with oxidative phosphorylation (Fig. 4B), whereas the upregulated genes were predominantly related to the spliceosome (Fig. 4C). The joint assessment of ATAC-seq and RNA-seq data identified two oxidative phosphorylation-related genes, COX6B1 (cytochrome c oxidase subunit 6B1) and NDUFA1 (NADH dehydrogenase (ubiquinone) alpha subcomplex-1). The integrative genomics viewer and RNA-seq data of COX6B1 and NDUFA1 were further analyzed (Fig. 4D).

Figure 4 Integration analysis of ATAC-seq and RNA-seq.

(A) Venn analysis of differential genes and peaks related genes. (B) KEGG pathway analysis of downregulated genes. The x-axis represents the generatio while y-axis represents the pathway. Dot sizes and colors represent the count numbers and adjusted p value. (C) KEGG pathway analysis of upregulated genes. The x-axis represents the generatio while y-axis represents the pathway. Dot sizes and colors represent the count numbers and adjusted p value. (D) Integrative genomics viewer (IGV) shows peaks located around oxidative phosphorylation pathway-related genes COX6B1 and NDUFA1. *P < 0.05, **P < 0.01, compared to the control Sc group; the unpaired t-test was used. FPKM: fragments per kilobase of exon model per million mapped fragments.

As depicted in Fig. 4D, the control treatment group exhibited three peaks for COX6B1 and two peaks for NDUFA1, while the ADSCP2 treatment group showed one peak for COX6B1 and none for NDUFA1. This suggests reduced accessibility of COX6B1 and NDUFA1 following ADSCP2 treatment. A subsequent analysis of the RNA-seq data revealed a downregulation of both COX6B1 and NDUFA1 in the ADSCP2 group. According to the UCSC genome browser analysis, the promoter regions of COX6B1 and NDUFA1 are characterized by a high prevalence of histone modifications (specifically, histone H3 lysine 27 acetylation, or H3K27Ac) and multiple transcription factor binding sites (Figs. 5, 6).

Figure 5 UCSC genome browser analysis of the promoter region of COX6B1.

(A) Whole gene analysis for COX6B1 including the chromosome location and promoter analysis. ENCODE candidate cis-regulatory elements (cCREs) combined from all cell types, cCREs are the subset of representative DNase hypersensitive sites across ENCODE and Roadmap Epigenomics samples that are supported by either histone modifications (H3K4me3 and H3K27ac) or CTCF-binding data. Red color represents the promoter-like signature, orange color represents proximal enhancer-like signature. H3K27Ac Mark, enrichment of the H3K27Ac histone mark across the genome as determined by a ChIP-seq assay, is thought to enhance transcription. (B) Promoter region analysis of identified peaks in the Control group for COX6B1. Predicted transcription factor binding sites using JASPAR (http://jaspar.genereg.net/). Blue box indicates the peaks’ chromosome location.

Figure 6 UCSC genome browser analysis of the promoter region of NDUFA1.

(A) Whole gene analysis for NDUFA1 including the chromosome location and promoter analysis. ENCODE candidate cis-regulatory elements (cCREs) combined from all cell types, cCREs are the subset of representative DNase hypersensitive sites across ENCODE and Roadmap Epigenomics samples that are supported by either histone modifications (H3K4me3 and H3K27ac) or CTCF-binding data. Red color represents the promoter-like signature, orange color represents proximal enhancer-like signature. H3K27Ac Mark, enrichment of the H3K27Ac histone mark across the genome as determined by a ChIP-seq assay, is thought to enhance transcription. (B) Promoter region analysis of identified peaks in the Control group for NDUFA1. Predicted transcription factor binding sites using JASPAR (http://jaspar.genereg.net/). Blue box indicates the peaks’ chromosome location.

Peak 1 is situated at the promoter region, while peaks 2 and 3 are found at the intron regions of COX6B1 (Fig. 5B). Similarly, for NDUFA1, peak 1 is located at the promoter region, and peak 2 is found at the intron region (Fig. 6B). The JASPAR transcription factor prediction analysis suggests that transcription factors such as ELK4 (ETS transcription factor), FLI1 (friend leukemia virus integration 1), ZNFs (zinc finger protein), and KLFs (Kruppel-like factors) may potentially bind with the COX6B1 promoter region, thereby influencing its transcription (Fig. 5B). Additionally, SP1 (specificity protein 1), KLFs, ZNFs, and SREBFs (sterol regulatory-element binding proteins) are likely to bind with NDUFA1 promoter regions (Fig. 6B). These findings show that ADSCP2 reduces the transcriptions of COX6B1 and NDUFA1 genes useful for the OXPHOS process. The mechanisms may be that ADSCP2 functions by regulating histone modifications or transcription factor binding sites on those promoter regions.

ADSCP2 inhibits the mRNA expression levels of COX6B1 and NDUFA1, as well as reduces cellular ATP and lactic acid levels

To investigate the impact of ADSCP2 on the transcription of COX6B1 and NDUFA1, we conducted quantitative PCR analysis. The results indicated a reduction in mRNA levels of both COX6B1 and NDUFA1 in the ADSCP2 treatment group (Fig. 7A). Additionally, we assessed ATP and lactic acid levels in cell pellets and culture medium following ADSCP2 treatment (Fig. 7B). The findings revealed a decrease in intracellular ATP levels, while ATP levels in the culture medium were elevated, albeit marginally, post-ADSCP2 treatment (Fig. 7C). Furthermore, intracellular lactic acid levels were significantly reduced, whereas lactic acid levels in the culture medium remained unchanged after ADSCP2 treatment (Fig. 7D).

Figure 7 ADSCP2 was found to inhibit the mRNA expression levels of COX6B1 and NDUFA1, as well as to reduce cellular ATP and lactic acid levels.

(A) Quantitative PCR analysis was conducted to assess the mRNA expression levels of COX6B1 and NDUFA1 following treatment with ADSCP2. (B) Graphical representation of the detection process for ATP and lactic acid levels in cell pellets and culture medium. (C) Measurement of ATP levels in cell pellets and culture medium, with normalization to protein content. (D) Measurement of lactic acid levels in cell pellets and culture medium, with normalization to protein content. Statistical significance is indicated as follows: * denotes P < 0.05; ** denotes P < 0.01.

Discussion

In an environment characterized by low energy, fibroblasts possess the ability to survive, albeit without the capacity to replicate or synthesize collagen. This research, grounded in bioinformatics analysis, presents an integrated analysis of ATAC-seq and RNA-seq in hypertrophic scar fibroblasts treated with ADSCP2. The findings of this study illuminate the regulatory role of ADSCP2 in the oxidative phosphorylation pathway within hypertrophic scar fibroblasts.

Oxidative phosphorylation, a critical metabolic pathway within mitochondria, liberates energy through the oxidation of nutrients such as carbohydrates, fats, and proteins into ATP. It is noteworthy that most normal cells predominantly rely on oxidative phosphorylation for the generation of ATP (Wilson, 2017).

The cessation of oxidative phosphorylation can potentially circumvent the generation of oxygen-free radicals, thereby averting apoptosis (Nolfi-Donegan, Braganza & Shiva, 2020). Sonic Hedgehog (SHH) has been observed to amplify oxidative phosphorylation in macrophages, bolster macrophage efferocytosis, and stimulate M2 polarization, which collectively contribute to the advancement of cutaneous scar formation (Zhang et al., 2024). Furthermore, the interruption of oxidative phosphorylation through the use of IM156 has been found to mitigate mitochondrial metabolic reprogramming and inhibit the development of pulmonary fibrosis (Willette et al., 2021). The suppression of miR-27b-3p has been observed to enhance mitochondrial oxidative phosphorylation and inhibit cardiomyocyte hypertrophy (Li et al., 2022). Concurrently, it appears that ADSCP2 may play a role in the inhibition of oxidative phosphorylation, subsequently leading to a reduction in scar formation.

The role of mitochondrial function in the survival and functionality of fibroblasts is crucial for cell repair, growth, and the production of the extracellular matrix during the wound healing process. It has been observed that the functioning of mitochondria fluctuates in response to alterations in the composition of the extracellular matrix, thereby regulating extracellular matrix. There is a differential regulation of collagen and fibronectin expression, as well as mitochondrial membrane potential, in normal skin and hypertrophic scar fibroblasts in response to the basic fibroblast growth factor (Song et al., 2011). Our research determined that ADSCP2 potentially influences mitochondrial function by targeting oxidative phosphorylation and two specific genes, COX6B1 and NDUFA1.

COX6B1, a component of the cytochrome c oxidase, is situated in the inner mitochondrial membrane and is integral to the final enzyme in the mitochondrial electron transport chain, which is responsible for driving oxidative phosphorylation. A deficiency in mitochondrial complex IV, which can be caused by mutations in COX6B1, has been linked with hydrocephalus, encephalomyopathy, and cardiomyopathy (Abdulhag et al., 2015).

NDUFA1, a mitochondrial complex protein situated in the mitochondrial inner membrane, contributes to the maintenance of the mitochondrial membrane potential and chemical proton gradient. This is achieved through oxidative phosphorylation via electron transport and ATP production. The mitochondrial membrane potential was found to be associated with the level of reactive oxygen species (ROS) (Lee et al., 2023). In the context of hypertrophic scarring, a significant increase in ROS levels was observed. In the recent study conducted by Yang et al. (2024), it was elucidated that drug-loaded microneedles have the potential to modify the pathological microenvironment of hypertrophic scar tissues in female rabbits through the process of ROS scavenging. Furthermore, this study revealed that ADSCP2 inhibited the transcription of COX6B1 and NDUFA1. Consequently, it is plausible that ADSCP2 could influence the mitochondrial membrane potential and ROS level by targeting NDUFA1.

The impact of ADSCP2 on these transcriptions for COX6B1 and NDUFA1 could be attributed to two potential mechanisms. The first mechanism suggests that the enrichment of histone H3 lysine 27 acetylation on the promoters of COX6B1 and NDUFA1 may be diminished in the group treated with ADSCP2.

Secondly, it is plausible that various DNA-binding transcription factors, including ETS (e-twenty six), zinc finger and kruppel-like transcription factors such as ELK4, FLI1, ZNFs, and KLFs, may bind to the COX6B1 promoter region. Additionally, SP1 (specificity protein 1), KLFs, ZNFs, and SREBFs (sterol regulatory-element binding proteins) could potentially bind to the NDUFA1 promoter regions. It is hypothesized that ADSCP2 may influence the binding of these transcription factors to the promoter regions, thereby reducing the transcriptions of COX6B1 and NDUFA1. Further research is required to elucidate this potential mechanism.

Conclusions

Peptides possess numerous distinctive benefits, including elevated biological activity, reduced immunogenicity, and robust specificity. Previously, we identified that ADSCP2 impacts pyruvate carboxylase, potentially influencing the oxidative phosphorylation pathway. Pyruvate carboxylase necessitates biotin and ATP to catalyse the carboxylation of pyruvate to oxaloacetate, playing a crucial role in gluconeogenesis and lipogenesis. In this study, an integrated analysis of ATAC-seq and RNA-seq demonstrates that ADSCP2 downregulates the oxidative phosphorylation pathway in hypertrophic scar fibroblasts. This research offers preliminary insights that could guide subsequent investigations into the peptide ADSCP2. Consequently, it may be viewed as a pioneering agent for the formulation of future strategies in the clinical treatment of scars.

Supplemental Information

Supplemental Information 1 Peak related genes in the ATAC-Seq for the Control group.

Supplemental Information 2 Peak related genes in the ATAC-seq for the ADSCP2 treatment group.

Supplemental Information 3 Differential peaks related genes in the ATAC-seq.

ADSCP2 compared to the Control group.

Additional Information and Declarations

Competing Interests

The authors declare that they have no competing interests.

Author Contributions

Qian Li conceived and designed the experiments, performed the experiments, analyzed the data, authored or reviewed drafts of the article, and approved the final draft.

Zhe Quan conceived and designed the experiments, performed the experiments, authored or reviewed drafts of the article, and approved the final draft.

Ling Chen performed the experiments, prepared figures and/or tables, and approved the final draft.

Yiliang Yin analyzed the data, prepared figures and/or tables, and approved the final draft.

Xin Chen analyzed the data, prepared figures and/or tables, authored or reviewed drafts of the article, and approved the final draft.

Jingyun Li analyzed the data, prepared figures and/or tables, authored or reviewed drafts of the article, and approved the final draft.

Human Ethics

The following information was supplied relating to ethical approvals (i.e., approving body and any reference numbers):

The Ethics Committee of the Women’s Hospital of Nanjing Medical University (Nanjing Women and Children’s Healthcare Hospital) approved this study (2022KY-162).

DNA Deposition

The following information was supplied regarding the deposition of DNA sequences:

The RNA-seq data pertaining to the impact of peptide ADSCP2 on the gene expression of primary hypertrophic scar fibroblasts is available at NCBI GEO: GSE202718.

The genes list for RNA-seq is available at Figshare: Li, Jingyun (2024). ADSCP2 RNA-seq all.counts.group2_vs_group1.edgeR_all.xls. figshare. Dataset. https://doi.org/10.6084/m9.figshare.26524723.v1.

Data Availability

The following information was supplied regarding data availability:

The raw ATAC-seq peak of Control, ADSCP2 and differential are available in the Supplemental Files.

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
