# Peer review of "Integrated analysis of ATAC-seq and RNA-seq reveals ADSCP2 regulates oxidative phosphorylation pathway in hypertrophic scar fibroblasts"

_PeerJ, doi:10.7717/peerj.18902_

## Round 0.1 · original submission · Major Revisions

Dear Dr. Li,

Thank you for the opportunity to analyse this manuscript on the role of ADSCP2 in regulating the oxidative phosphorylation (OXPHOS) pathway in hypertrophic scar fibroblasts via ATAC-Seq and RNA-Seq integration. The manuscript presents interesting insights into ADSCP2’s potential role in scar formation and fibrosis. However, the study as presented raises significant concerns regarding methodology, hypothesis clarity, data interpretation, and supporting evidence. Below are detailed suggestions to address these issues and strengthen the manuscript. Please provide a detailed reply to each reviewer´s comments.

Major Points for Revision
Hypothesis Clarity and Background Context:

The manuscript would benefit from a clearer hypothesis. While the authors refer to ADSCP2's influence on hypertrophic scar fibroblasts, this connection lacks clear, mechanistic grounding. An explicit hypothesis would guide readers in understanding the rationale behind each experimental step.
ADSCP2’s background, especially any previous findings, should be better articulated. Without prior understanding of ADSCP2’s role, readers may struggle to grasp its relevance to scar formation.

Recommendation Summary
This manuscript has the potential to contribute valuable insights into hypertrophic scar fibroblast metabolism. However, major revisions are required to clarify the hypothesis, improve methodological transparency, and provide experimental support for the conclusions. Adding functional assays or reducing interpretive claims could significantly increase the manuscript's robustness and impact in the field.

Reviewer 1 ·

Basic reporting

Title: The authors should make it abundantly clear ADSCP2 is the name they selected for peptide fragment of ALCAM. Additionally, should it be “Integrated analysis” rather than “Integration analysis”?

The introduction should focus on introducing ADSCP2, the authors’ previous findings, and what their new experiments contribute to expand the knowledge of the field. The introduction could be improved as following:

In lines 60 – 62 the authors allude to previous work which denoted ADSPC2 as key peptide regulating the fibrotic response in hypertrophic scars. A better use of their introduction would be to explain to the readers what was performed and observed previously, in a succinct manner. As it stands, the authors leave the onus on the readers to refer back to their previous study to gain any inclination on why ADSPC2 is significant in terms of hypertrophic scar formation. In line 62- 65, the authors offer many hypothetical events, but no concrete statements of a hypothesis. Much of this language could be saved for the discussion.

Line 195 defines KEGG, although this is not the first time the authors use the abbreviation in their manuscript.

Experimental design

Results: The wording in lines 164 – 166 of the GO assessment of the author’s ATACseq results is very vague, giving little significance to the work performed. A more thorough assessment of their ARC associated genes could be performed.

In lines 184 through 189, it is unclear if the authors have used new samples or re-analyzing their previously published data. If the later, they need to explicitly say as much, as their methods make it appear that they are performing cell isolation and RNAseq unique to this manuscript.

In lines 205 – 208, the authors select two of the “oxidative phosphorylation-related genes” from their joint assessment of ATAC and RNAseq datasets. Why were these genes chosen out of all the oxidative phosphorylation-related genes? As their manuscript is worded, it appears like they could be chosen at random.

The authors leverage their joint dataset analysis heavily in this manuscript, but lack any supporting experiments that ultimately prove whether or not their peptide regulates fibroblast oxidative phosphorylation in their in vitro model.

Validity of the findings

All assessment on NDUFA1 and COX6B1 are performed in silico, with no definitive support to their function in terms of hypertrophic scar formation. Additional data is needed to make any claim regarding their role in oxidative phosphorylation in the context of hypertophic scar associated fibroblasts. This is the only data supporting the authors’ title that ADSCP2 regulates oxidative phosphorylation in hypertrophic scar associated fibroblasts and thus needs to be more robust to make such claims.

Reviewer 2 ·

Basic reporting

Needs to improve

Experimental design

Experiment properly designed

Validity of the findings

Issues to address

Additional comments

See attached

Annotated reviews are not available for download in order to protect the identity of reviewers who chose to remain anonymous.

---

## Round 0.2 · accepted · Accept

Dear Dr. Li,

Congratulations on the acceptance of your manuscript.

Reviewer 2 ·

Basic reporting

I am satisfied with the revision and basic reporting is in order.

Experimental design

All good and within Aims and Scope of the journal.

Validity of the findings

Very informative and adds value to the already existing literature.

Additional comments

I would like to take this opportunity to thank the authors for an engaging response and for addressing most if not all the worries I mentioned.

The changes have greatly improved the readability of the manuscript and enhance the quality of the work.